# Single nucleotide polymorphism leads to daptomycin resistance causing amino acid substitution—T345I in MprF of clinically isolated MRSA strains

**Masaki Nakamura**[1,2,3]*, **Hayato Kawada**[2], **Hiroki Uchida**[1], **Yusuke Takagi**[1], **Shuichi Obata**[4,5], **Ryotaro Eda**[2], **Hideaki Hanaki**[3], **Hidero Kitasato**[1,2]

1 Department of Microbiology, Kitasato University School of Allied Health Sciences, Kanagawa, Japan, 2 Department of Environmental Microbiology, Kitasato University Graduate School of Medical Sciences, Kanagawa, Japan, 3 Research Center for Infection control, Kitasato Institute for Life Sciences, Kitasato University, Tokyo, Japan, 4 Department of Anatomical Sciences, Kitasato University School of Allied Health Sciences, Kanagawa, Japan, 5 Department of Histology and Cell Biology, Yokohama City University School of Medicine, Kanagawa, Japan

* m-nakamu@kitasato-u.ac.jp

**Data Availability Statement:** All relevant data are within the manuscript and its Supporting Information files.

## Abstract

Daptomycin (DAP) is one of the most potent antibiotics used for the treatment of methicillin-resistant *Staphylococcus aureus* (MRSA) infections. Due to an increase in its administration for combating MRSA infections, DAP non-susceptible (DAP-NS) MRSA strains have recently been reported in clinical settings. The presence of single nucleotide polymorphisms (SNPs) in the multiple peptide resistance factor (*mprF*) gene is the most frequently reported cause for the evolution of DAP-NS MRSA strains; however, there are some variations of SNPs that could lead to DAP-NS. In this study, we used two clinical MRSA strains, including DAP susceptible (DAP-S) and DAP-NS, isolated from the same patient at different time points. We introduced T345I SNP to *mprF* of the DAP-S MRSA strain using the gene exchange method with pIMAY vector. Further, we investigated the phenotype of the mutant strain, including drug susceptibility, cell surface positive charge, and growth speed. The mutant strain exhibited (i) resistance to DAP, (ii) up-regulation of positive surface charge, (iii) slower growth speed, and (iv) thickened cell walls. Hence, the SNP in *mprF* may have caused an up-regulation in MprF function, with a subsequent increase in positive surface charge. Cumulatively, these results demonstrated that the T345I amino acid substitution in *mprF* represents one of the primary causes of DAP-NS in MRSA strains.

## Introduction

Methicillin-resistant *Staphylococcus aureus* (MRSA) is a major cause of hospital-acquired and community-acquired infections. MRSA shows resistance to structurally and functionally diverse antibiotics; thus, reducing the options for suitable and effective therapeutic agents. Vancomycin (VAN) is widely used to combat drug resistant bacteria and MRSA; however, due

**Funding:** This study was supported by a grant from Kitasato University School of Allied Health Sciences (Grant-in-Aid for Research Project, No. 2016-1052).

**Competing interests:** The authors have declared that no competing interests exist.

to the emergence of VAN resistant clinical isolates, including VAN intermediate-resistant MRSAs (VISA), heterogeneous-resistant VISA (hVISA), and ß-lactam induced VAN resistant MRSA (BIVR), the use of VAN has declined [1]. After the clinical usage of VAN began, the following new anti-MRSA agents have launched: teicoplanin (TEC), linezolid (LZD), daptomycin (DAP), and tedizolid (TZD); however, effective potent anti-MRSA therapeutic agents remain limited. Although DAP plus beta-lactam combination therapy is becoming increasingly popular, insufficient data has been provided from clinical trials to demonstrate its efficacy [2].

DAP, a uniquely structured antibiotic consisting of a cyclic lipopeptide, exhibits a potent and rapid bactericidal activity against a wide variety of gram-positive bacteria, including VAN resistant *Staphylococcus* and *Enterococcus* [3]. Its mode of action involves integration of a calcium-DAP complex into the cytoplasmic membrane forming ion-permeable transmembrane channels, followed by dissipation of membrane potential, and ultimately causing cell rupture. Clinical use of DAP was approved in 2003 and 2011 in the US and Japan, respectively. Since this time, its use has increased; however, frequent use of DAP in clinical settings has caused the emergence of DAP-resistant clinical isolates, including *Staphylococcus* and *Enterococcus* [1, 4]. According to the CLSI criteria, *Staphylococcus aureus* (*S. aureus*) with MIC of DAP $\leq 1$ μg/mL and $> 1$ μg/mL is categorized as DAP susceptible (DAP-S) and DAP non-susceptible (DAP-NS) strains, respectively [5].

Although DAP-NS MRSA strains are uncommon, several DAP-NS MRSA isolates have been reported in clinical settings across the world [4, 6–9], which were found to contain single point mutations in their genes resulting in amino acid substitution [10, 11]. The DAP-NS MRSA proteins reported to harbor frequent mutations are as follows: multiple peptide resistance factor (MprF), a bifunctional protein for lysylphosphatidylglycerol (L-PG) synthesis and L-PG flippase; YycG, a histidine kinase; RpoB and RpoC, the subunits of RNA polymerase [12]. Among them, single nucleotide polymorphisms (SNPs) in the *mprF* gene (T345I, S295L, and S337L [13]) have been reported most frequently. Specifically, SNPs in *mprF* result in a gain-of-function in L-PG synthesis from lysyl-tRNA and membrane phosphatidylglycerol (PG) [14]. The resulting L-PG is a positively charged phospholipid and is translocated to the outer leaflet of the cytoplasmic membrane, where it increases the net positive charge of the bacterial cell surface [15, 16].

Various mechanisms have been postulated to confer DAP resistance: (i) L-PG increases the net positive charge on the cell surface, which increases electrostatic repulsion of positively charged DAP molecules resulting in a low level of DAP intercalation into the membrane; (ii) mutations in the *walKR* two-component system regulate the synthesis of cell wall proteins followed by co-resistance to VAN and DAP; (iii) a single point mutation in *rpoB* induces DAP resistance via cell wall thickening and reduced cell surface negative charge [17–20].

To date, several genes associated with the development of DAP resistance in MRSA strains, as well as the possible consequences of mutations in these genes, have been reported. However, it remains unclear whether variants of the MprF protein are capable of independently evoking the onset of DAP resistant mechanisms. To address this issue, we cloned a mutant *mprF* gene from DAP-NS MRSA clinical isolates; the mutation was then transferred back to DAP-S MRSA strains isolated from the same patient. The results demonstrated that the *mprF* mutation increased the MIC of DAP from the level of the DAP-S MRSA strain to the DAP-NS MRSA strain.

## Materials and methods

### Bacterial strains

Two clinical MRSA strains (KAM17 and KAM18; Table 1) were used in this study, both of which were isolated from the same patient, who was admitted to a hospital in Japan and

**Table 1. Bacterial strains and plasmids used in this study.**

| Strains/Plasmids | Description | Source or Reference |
|---|---|---|
| **Strains** | | |
| KAM17 | DAP-susceptible MRSA | Clinical isolate |
| KAM18 | DAP-non-susceptible MRSA | Clinical isolate |
| KAM17mut | KAM17 transduced with MprF T345I | This study |
| KAM17mutR | Reverse mutant from KAM17mut | This study |
| **Plasmids** | | |
| pMD20-T | Cloning vector for TA-cloning | TaKaRa bio Inc. |
| pIMAY | Shuttle vector for gene replacement in Gram-positive bacteria | [21] |
| pIMAY-*mprF*-WT | pIMAY vector containing wild-type *mprF* gene | This study |
| pIMAY-*mprF*-T345I | pIMAY vector containing mutant *mprF* gene (T345I) | This study |

treated with VAN followed by DAP. KAM17 was isolated before, and KAM18 was isolated after DAP administration. Antibiotic susceptibility testing, performed by the hospital, determined that KAM17 was susceptible to DAP, while KAM18 was non-susceptible.

## Genotyping of isolated strains with POT method

The genotyping of each strain was performed by the PCR-based ORF Typing (POT) method, which is a molecular epidemiological method capable of identifying and distinguishing isolates at the strain-level [22]. We identified these strains with the Cica Geneus® Staph POT KIT (Kanto Chemical, Japan) according to the manufacturer's manual.

## Sequencing analysis of *mprF*

Genomic DNA was isolated from each MRSA strain as follows. Bacteria were cultured on trypticase soy agar at 37˚C overnight. Bacterial colonies were then scraped from the agar plate, resuspended in 100 μL of lysis buffer (20 μg/mL lysostaphin, 1000 U/mL achromopeptidase in TE buffer), and incubated for 10 min at 37˚C. Next, 50 μL of 0.5M KOH and 50 μL of 1M Tris/HCl were added for alkaline treatment and neutralization, respectively. The obtained solution contained DNA, which was used in subsequent experiments. Using 2 μL of the DNA aliquot as a template, PCR amplification of *mprF* was performed by TaKaRa Taq HS Perfect Mix (Takara Bio, Japan) with the primers listed in Table 2. PCR clean-up was performed by ExoSAP-IT (Thermo Fisher Scientific, MA, USA), and the final product was used as a sequencing template. Sequence analysis was performed by Eurofins Genomics (Tokyo, Japan) with the primers listed in Table 2. The sequencing data were analyzed using ApE software (created and maintained by Wayne Davis from the University of Utah) and aligned with the reference sequence of *mprF* of *Staphylococcus aureus* N315 (GenBank accession number NC_002745.2).

## Genetic exchange of mutant *mprF* with pIMAY vector

Plasmids used for genetic exchange in this study are shown in Table 1. The wild-type *mprF* gene (*mprF*-WT) in KAM17 and the mutant *mprF* gene (*mprF*-T345I) in KAM18 were cloned in a pIMAY vector as follows: *mprF* from each strain was amplified by PCR using Mighty TA-cloning Kit (Takara). The restriction enzyme sites for ClaI and NotI were attached to the PCR primers (Table 2). The PCR products were cloned into the pMD20-T vector by TA-cloning. The *mprF* gene cloned in pMD20-T was spliced out by digestion with ClaI and NotI, transferred into the MCS of pIMAY vector, and designated as pIMAY-*mprF*-WT and pIMAY-*mprF*-T345I (Table 1).

**Table 2. Primers for sequencing and cloning of *mprF* gene.**

| Primers | Sequence (5'-3') |
| --- | --- |
| ***mprF* PCR Primers** | |
| *mprF*-F1 | GCACTCATAATCGGCTGTT |
| *mprF*-R1 | TTGGGCTGATAATAAAAGTT |
| ***mprF* Sequence Primers** | |
| *mprF*-F2 | ACCATATTGTTCTGTTTGAG |
| *mprF*-F3 | TATTGGTGCAGGCGTTAGAG |
| *mprF*-F4 | GGCGCTTTCGATTTAGTTGT |
| *mprF*-F5 | AGCTATTATTTTTGTTCTGC |
| *mprF*-F6 | TTTAACGCAATTTTCAACTT |
| *mprF*-R2 | TAAATCTAACTCTGGCAACC |
| *mprF*-R3 | TCCAAGAACCACTAATGAAC |
| *mprF*-R4 | GATAATGGCACGTCTACTTT |
| *mprF*-R5 | AGCGTCAACAATTACACCAC |
| *mprF*-R6 | CCACCAATAAATAGTAACAC |
| ***mprF* Cloning Primers** | |
| ClaI-*mprF*-F0 | AAAAAATCGATATGAATCAGGAAGTTAAAAACAAAA |
| NotI-*mprF*-R0 | AAAAAGCGGCCGCTTATTTGTGACGTATTACACGCATT |
| pIMAY-F | TACATGTCAAGAATAAACTGCCAAAG |
| pIMAY-R | AATACCTGTGACGGAAGATCACTTCG |

pIMAY (a kind gift from Dr. Foster of Trinity College, Dublin) is a shuttle vector for the genetic exchange of *Staphylococcus*. The plasmid contains temperature-sensitive gram-positive replication genes and a tetracycline-inducible antisense *secY* region, which enables exchange of arbitrary genes in *Staphylococcus* [21]. For genetic exchange from wild-type *mprF* to mutant *mprF* in KAM17, electrocompetent KAM17 cells were prepared according to the instructions provided for Gene Pulser Xcell™ (Bio-Rad).

The pIMAY-*mprF*-T345I construct was transformed into KAM17 competent cells by electroporation using Gene Pulser Xcell™ Electroporation System (Bio-Rad) according to the manufacturer's instructions. KAM17 harboring the plasmid were selected following incubation on trypticase soy agar containing 10 μg/mL chloramphenicol at 28° C for 48 h. After selection, KAM17 cells containing pIMAY-*mprF*-T345I were incubated overnight at 39°C for allelic exchange between genomic *mprF* (wild-type in KAM17) and plasmid *mprF* (mutant in pIMAY). Following recombination, plasmid excision was done as follows: For pre-incubation, KAM17 containing pIMAY-*mprF*-T345I was incubated overnight in trypticase soy broth at 28°C under shaking conditions. The broth was then diluted ten-fold and plated onto trypticase soy agar containing 1 μg/mL anhydrotetracycline. Plates were incubated at 28°C for 48 h for plasmid excision. To confirm plasmid excision, the developed colonies were streaked onto trypticase soy agar plates containing 10μg/mL chloramphenicol at 28°C for 48 h. A chloramphenicol susceptible clone was considered as a plasmid excised clone and designated as KAM17mut, which contained the mutant *mprF* gene. To construct a reverse mutant of KAM17mut, the pIMAY-*mprF*-WT plasmid was transferred into KAM17mut, as described above. The acquired reverse mutant was designated as KAM17mutR, which contained the wild-type *mprF* gene.

## Antibiotics susceptibility test

E-test determined the MIC of anti-MRSA agents against each strain. Overnight culture of each strain on trypticase soy agar was suspended in normal saline and adjusted to an $OD_{590}$ of 0.30.

We then collected 100 μL of the suspension and plated it on Mueller-Hinton E agar (Sysmex bioMérieux, France) using a sterilized swab. Then E-test strips of DAP, VAN, TEC, LZD, and oxacillin (OXA) (Sysmex bioMérieux) were placed on the agar. After 24-h- incubation at 37°C, the MIC of each antimicrobial agent was measured, which was performed three times, and the median value of the three experiments was defined as the MIC of the antibiotic against the strain.

## Population analysis for daptomycin

Population analysis of each strain was performed according to a previously described method [23]. This analysis shows the drug resistant level of a strain subpopulation. Strains were cultured overnight in trypticase soy broth, and subsequently diluted to an $OD_{590}$ of 0.30. This dilution was serially diluted by ten-fold, and 100 μL of each dilution was spread on Mueller-Hinton agar (Becton Dickinson, NJ, USA) containing various concentrations of DAP (0 μg/mL to 5 μg/mL). Calcium concentration in the agar was adjusted to 50 μg/mL. Plates were incubated for 48 h at 37°C, and the colonies were counted. The experiment was performed twice.

## Cytochrome c binding assay

Bacteria were cultured on trypticase soy agar for 18 h at 37°C, then washed with 20 mM MOPS buffer twice. The bacterial cells were harvested by centrifugation at $19,000 \times g$ for 2 min, and the pellet was resuspended in 20 mM MOPS buffer. Bacterial density was adjusted to an $OD_{590}$ of 7.0 in the MOPS buffer. An aliquot of 200 μL of this suspension was then transferred into a new 2 mL tube. Cytochrome c (Sigma) solution (2.5 mg/mL in MOPS buffer) was added to the bacterial suspension (final concentration 0.5 mg/mL) and incubated for 10 min at room temperature. The solution was centrifuged for 2 min at $19,000 \times g$, and the supernatant was harvested and filtrated with a 0.2 μm pore size filter (Minisart RC4, Sartorius, Germany). The filtrate was assayed photometrically at 530 nm with Gene Quant 100 (GE Healthcare, IL, USA). Six independent runs were performed.

## Growth curve and doubling time

Overnight culture of each strain in trypticase soy broth was diluted 1000 times in fresh trypticase soy broth. The diluted broth was incubated at 37°C with shaking for 48 h, and $OD_{660}$ of the broth was measured at an interval of 2 min by absorptiometer (TVS062CA: Advantec, Tokyo, Japan). Doubling time during the exponential growth phase was calculated by the regression line of the growth curve. Three independent runs were performed on separate days.

## Transmission electron microscopy

Cell wall thickness was evaluated by transmission electron microscopy as described previously [24–26]. Bacteria were cultured in trypticase soy broth for 18 h at 37°C, then diluted 100 times by fresh trypticase soy broth and cultured for 6 h at 37°C to a logarithmic growth phase. The cells were harvested and washed twice with phosphate-buffered saline, then fixed in 2% glutaraldehyde in 0.1 M phosphate-buffered saline for 2 h at room temperature, and in 1% osmium tetroxide in the same buffer for 1 h on ice. Fixed cells were then dehydrated in a graded series of ethanol and propylene oxide, and subsequently embedded in EPON812 (TAAB, UK). Ultra-thin sections were cut and stained with uranyl acetate and lead citrate, and photographic images were obtained at a final magnification of 20,000× with a transmission electron microscope (H7500, Hitachi High-Technologies, Japan) equipped with a digital camera system. The

thicknesses of the cell wall with nearly equatorially cut surfaces were measured. The thicknesses of each cell was calculated as follows: a transparent grid made up of 20 radial lines arranged regularly at angles of 18° was placed on the center of each cell. The interaction zones between the grid line and the cell wall were then measured at ten different points, the mean of which was considered to designate the cell thicknesses. Fifty cells in each strain were measured; the results were expressed as mean ± SDs.

## Statistical analysis

Statistical analysis was performed using KaleidaGraph (Synergy Software, PA, USA). ANOVA with Tukey-Kramer post-test was used to compare results between more than two groups.

## Results

### POT typing of each strain

The genotyping of each strain was carried out by the POT method (Fig 1). Each strain showed the same band pattern, and the POT type was 92-244-113, indicating MRSA SCC*mec* type II. This result showed that the two clinical isolates were derived from the same origin.

### Point mutation of *mprF* in DAP-NS strain

The sequence analysis of *mprF* in each strain is shown in Fig 2. The 1034th cytosine was replaced with thymine in KAM18, a DAP-NS MRSA strain (*Staphylococcus aureus* N315 was taken as the reference strain). This SNP resulted in a missense mutation in *mprF*, which led to T345I amino acid substitution in MprF. DAP-S MRSA strain KAM17 had no mutation in the *mprF* gene.

### Correlation of DAP susceptibility with *mprF* mutation

Antibiotics susceptibility of each strain is shown in Table 3. KAM17 isolated before administration of DAP was susceptible to DAP (MIC ≤ 1 μg/mL); KAM18 isolated after administration of DAP was non-susceptible to DAP (MIC > 1μg/mL). The constructed mutant MprF (T345I) strain KAM17mut showed non-susceptibility to DAP, while the reverse mutant strain KAM17mutR was susceptible to DAP. In *mprF* mutant strains, MICs of VAN and TEC were elevated, whereas MIC for LZD was reduced. The MICs of OXA were >256 μg/mL in all strains.

### Population analysis in the presence of DAP

To investigate the susceptibility of the mutant strain KAM17mut to DAP, population analysis of the strain in the presence of DAP was carried out. KAM17mut exhibited a DAP non-susceptible pattern similar to that of KAM18. Meanwhile, the parent strain KAM17 and reverse mutant strain KAM17mutR both exhibited a DAP susceptible pattern (Fig 3).

### Cell surface positive charge

Cytochrome c binding assay was carried out to measure the cell surface positive charge. After cytochrome c bound to the cell, the adsorbed abundance was calculated. Cytochrome c has a positive charge; thus, a more positively charged cell indicates the presence of more cytochrome c in the supernatant. Compared with KAM17 and KAMmutR, cytochrome c adsorption was significantly decreased in the *mprF* mutant strain KAM17mut to the same level as that observed for the DAP-NS strain of KAM18 (Fig 4). This result demonstrated that the surface of the *mprF* mutant strain had a higher positive charge than that of the wild-type strain.

**Fig 1. POT analysis for genotyping of each strain.** Clinical isolates KAM17 and KAM18 had the same POT pattern. Manipulated strains KAM17mut and KAM17mutR also had the same POT pattern as KAM17. The band of 355 bp in mixture 1 was SCC*mec* type II specific.

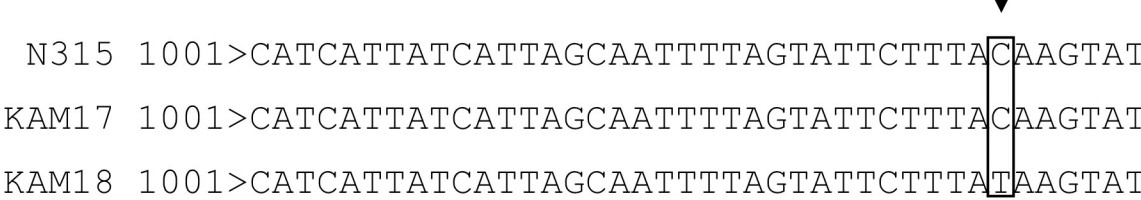

```
                                                           1034th
                                                             ↓
  N315  1001>CATCATTATCATTAGCAATTTTAGTATTCTTTACAAGTAT

KAM17  1001>CATCATTATCATTAGCAATTTTAGTATTCTTTACAAGTAT

KAM18  1001>CATCATTATCATTAGCAATTTTAGTATTCTTTATAAGTAT
```

**Fig 2. Sequence analysis of the *mprF* gene in each strain.** The 1034th cytosine was changed to thymine in KAM18 causing a missense mutation, which resulted in T345I amino acid substitution. N315 was used as a reference sequence of *mprF*.

**Table 3. MICs of each strain by E-test.**

| Bacterial strains | MIC of Antibiotics (µg/mL) | | | | |
|---|---|---|---|---|---|
| | **DAP** | **VAN** | **TEC** | **LZD** | **OXA** |
| KAM17 | 0.5 | 1.5 | 1.5 | 1.5 | >256 |
| KAM18 | 3 | 3 | 2 | 1.0 | >256 |
| KAM17mut | 3 | 3 | 2 | 1.0 | >256 |
| KAM17mutR | 0.5 | 1.5 | 1.5 | 1.5 | >256 |

Each value represents the median of three measurements.

DAP: daptomycin, VAN: vancomycin, TEC: teicoplanin, LZD: linezolid, OXA: oxacillin

## Growth curve and doubling time

To investigate whether *mprF* mutation affects cell growth speed, each strain was cultured in trypticase soy broth, and optical density was measured intermittently. The *mprF* mutant strain KAM17mut grew slower than the *mprF* wild-type strain KAM17 and KAM17mutR. Specifically, the doubling time of KAM17mut was 36.2 min, while that of KAM17 and KAMmutR was 27.8 min. The clinically isolated DAP-NS strain KAM18 also grew slower than DAP-S strains (Fig 5). These results indicate that the *mprF* mutation affected cell growth speed.

## Cell wall thickness

Considering that a thicker cell wall is a feature associated with DAP-NS strain [26], we also investigated the cell wall thickness of each strain by transmission electron microscopy. Results show that the cell wall in the DAP-NS strains KAM18 and KAM17mut were thicker than that of DAP-S strains KAM17 and KAM17mutR (Fig 6).

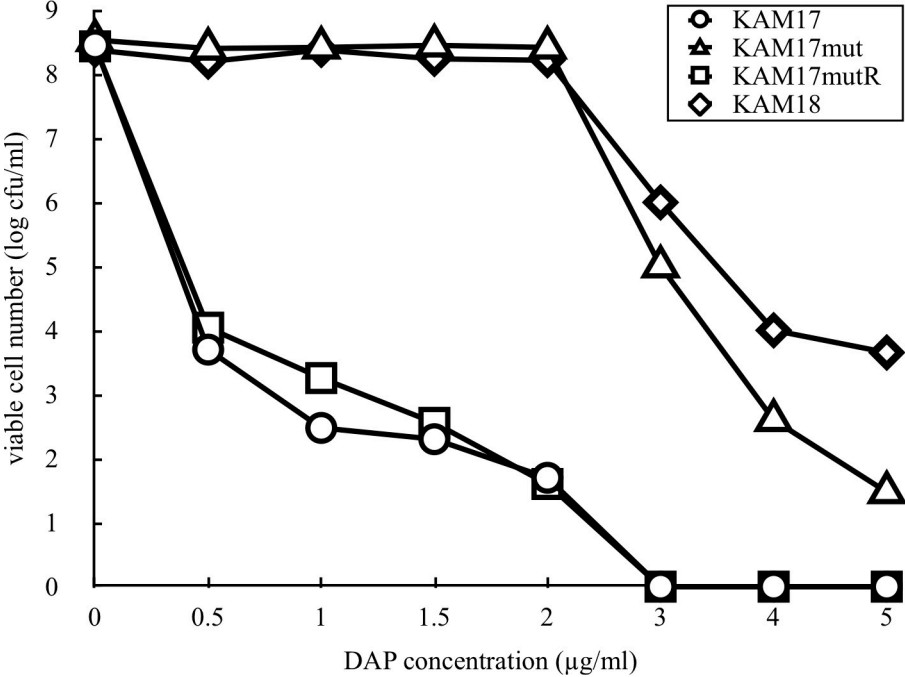

**Fig 3. Population analysis of each strain for DAP susceptibility.** The number of colonies developed on the plates containing various concentrations of DAP was counted. The data shown are representative of two independent experiments with similar results.

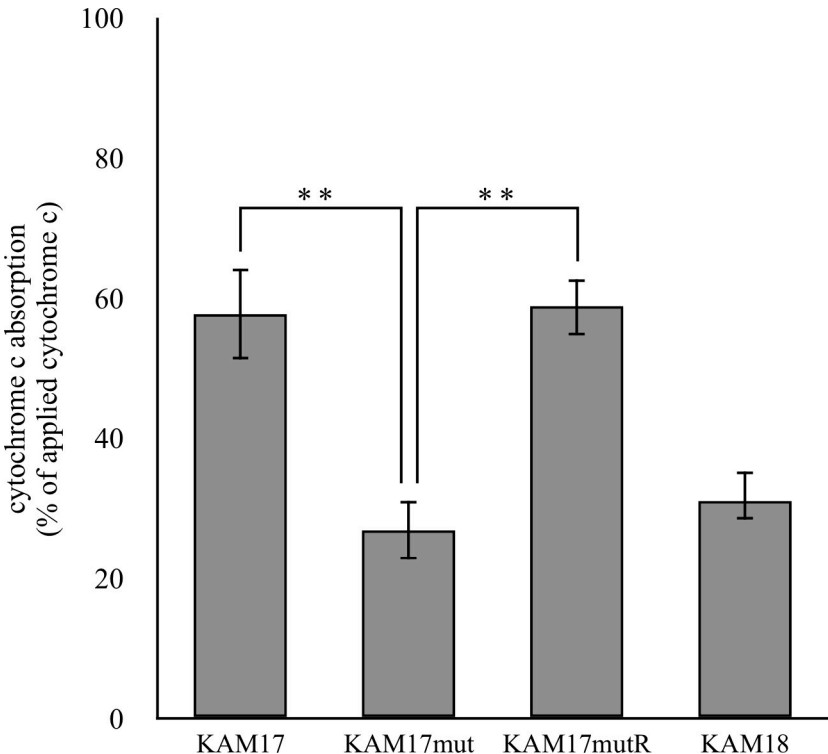

**Fig 4. Cytochrome c binding assay of each strain.** Each strain was incubated with cytochrome c. The adsorbed cytochrome c was calculated by the residual cytochrome c in the supernatant. The error bars indicate standard error from six independent experiments (** $p < 0.01$).

## Discussion

Although *mprF* mutations have been detected in DAP-NS MRSA strains [13], few studies have demonstrated whether the point mutations of *mprF* cause resistance against DAP in MRSA strains. A previous study showed that the L341F amino acid substitution in MprF elevated the MIC of DAP [26]; however, other amino acid substitutions in MprF caused by point mutations, including T345I, have not been confirmed as the cause of DAP resistance. In the present study, we demonstrated, using genetic exchange method with the pIMAY vector, that the introduction of T345I mutation into the *mprF* gene led to DAP resistance in MRSA strains.

The series of clinically isolated MRSA strains, KAM17 and KAM18, had the same genetic origin, as confirmed by the POT test. Moreover, the E-test showed that KAM17 was susceptible to DAP, while KAM18 was non-susceptible. From these results, it was speculated that DAP administration could induce DAP resistance. Previous studies have shown that *mprF* is involved in developing DAP resistance in MRSA strains [12]; thus, we performed sequence analysis for the *mprF* gene in these strains, which revealed that the DAP-NS strain KAM18 had an SNP in the *mprF* gene. Other DAP-NS MRSA strains have also been reported to contain an SNP in the *mprF* gene, which led to a T345I amino acid substitution [13]. Hence, the T345I amino acid substitution in MprF seemed to be one of the causes of DAP resistance in MRSA strains. To validate this hypothesis, we carried out the *mprF* exchange in a DAP-S MRSA strain by using pIMAY vector.

Genetic manipulation can directly demonstrate correlation between genetic mutations and phenotypes. The pIMAY vector is used for gene exchange in gram-positive bacteria, and can introduce point mutations in an arbitrary position of any gene. We, therefore, employed this

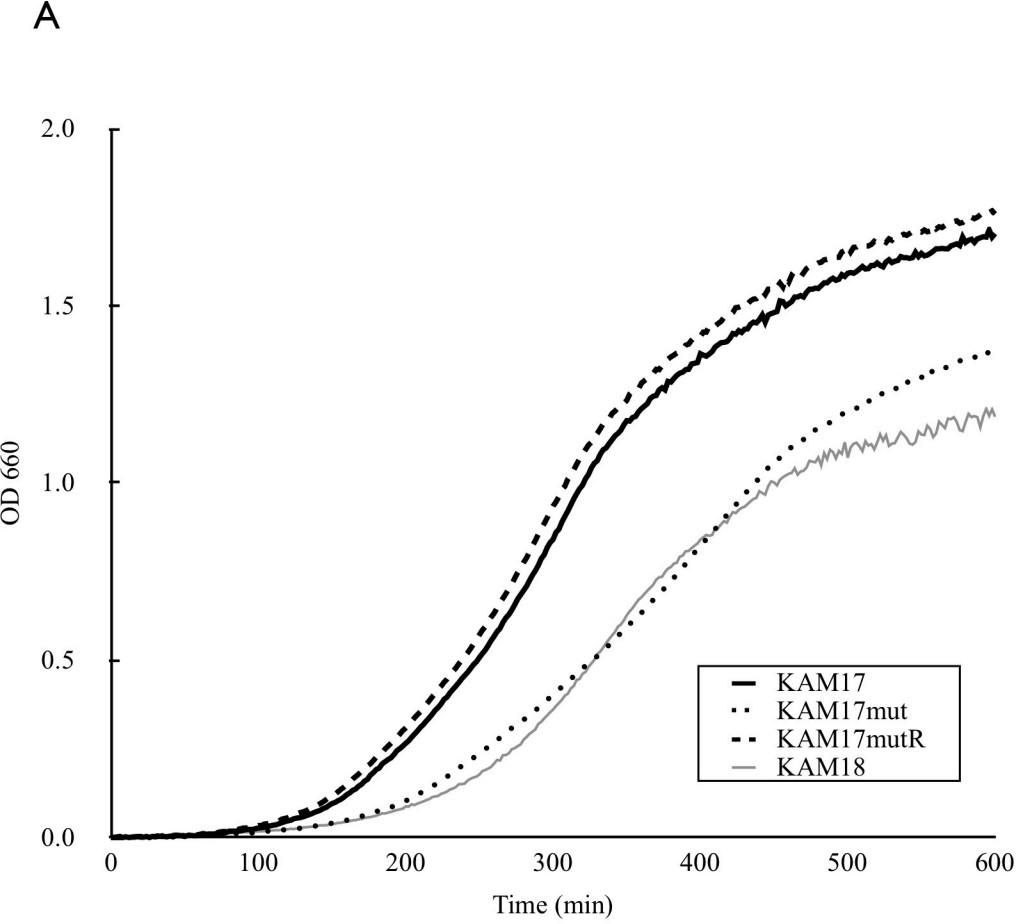

**A**

**B**

| Bacterial strains | Doubling time (min) |
|---|---|
| KAM17 | 27.8 ± 1.42 |
| KAM17mut | 36.2 ± 0.82** |
| KAM17mutR | 27.8 ± 1.36 |
| KAM18 | 42.2 ± 1.37** |

mean ± SD

**Fig 5. Growth curve of each strain.** (A) The concentration of bacteria was measured as the optical density at 660 nm. MprF mutant DAP resistant strain KAM18 and KAM17mut grew slower than DAP susceptible strains, KAM17 and KAM17mutR. The growth curves are representative of three independent experiments with similar results. (B) The doubling time of each strain was calculated according to its regression curve. The calculated values represent mean ± SD of three independent experiments (**$p < 0.01$ vs. KAM17).

system to introduce an SNP into the *mprF* gene of KAM17 (MprF wild-type) strain and obtained the mutant strain KAM17mut (MprF T345I). This genetically manipulated strain showed low susceptibility against DAP in a pattern that was similar to that of KAM18 (MprF T345I). To reinforce the relationship between the SNP and DAP-NS, we constructed a reverse

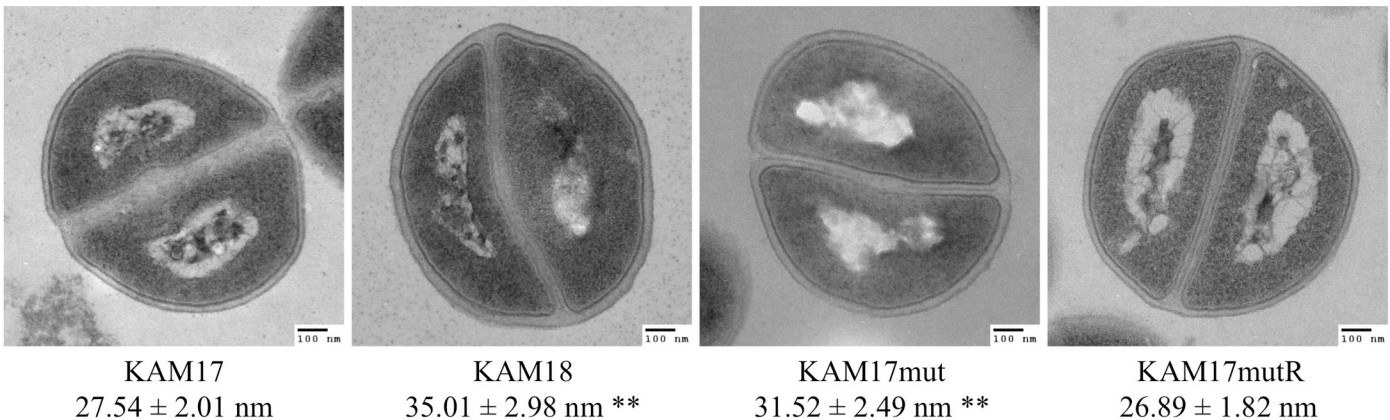

KAM17
27.54 ± 2.01 nm

KAM18
35.01 ± 2.98 nm **

KAM17mut
31.52 ± 2.49 nm **

KAM17mutR
26.89 ± 1.82 nm

**Fig 6. Transmission electron microscopy images of each strain.** (A) Cell wall thickness was measured at 200,000× magnification. Fifty measurements were obtained, and the results are expressed as mean ± SDs (**p<0.01 vs. KAM17).

mutant strain KAM17mutR derived from KAM17mut with the exchange from MprF T345I to wild-type MprF, using the pIMAY vector. The KAM17mutR (MprF wild-type) strain exhibited the same antibiotic susceptibility-pattern as KAM17 (MprF wild-type) in the E-test and population analysis. These results strongly support the hypothesis that the SNP in *mprF* causes DAP resistance in MRSA strains.

The mechanism of DAP-NS induced by *mprF* mutation remains unclear. MprF has two components, the lysinylation domain and the translocation domain [13]. The lysinylation domain adds lysine residues onto PG to form L-PG, while the translocation domain translocates L-PG from the membrane of the inner leaflet to the outer leaflet. The position of T345I amino acid substitution in MprF is located on both domains; thus, it is speculated that T345I amino acid substitution affects the membrane surface charge. Therefore, a putative mechanism for the DAP-NS MRSA strain is that gain-of-function mutation in *mprF* increases L-PG content in the outer membrane, followed by an up-regulation of membrane positive charge [17]. As a result, a positive membrane charge causes repulsion against DAP, making it more difficult for DAP to bind to the membrane. In this study, we evaluated cell surface positive charge in each strain by cytochrome c binding assay. The DAP-NS strain, KAM17mut bound less with cytochrome c compared with the DAP-S strains, KAM17 and KAM17mutR, suggesting that positive charge in the cell membrane was increased in DAP-NS strains. These data suggest that T345I amino acid substitution in MprF caused the gain-of-function of MprF. As a result, a positive membrane charge was up-regulated, followed by DAP repulsion.

We also observed that the colony size varied in the different strains with KAM17mut (DAP-NS) forming a smaller colony than KAM17 (DAP-S) or KAM17mutR (DAP-S; S3 Fig). We therefore evaluated the growth speed of these strains. KAM17mut (DAP-NS) was found to grow slower than KAM17 (DAP-S) and KAM17mutR (DAP-S), indicating that *mprF* mutation affected the growth speed of MRSA strain. These results agree with those of past studies, which reported that the growth rate of DAP-NS *mprF* mutants was lower than that of the DAP-S strain [27]. Although the mechanism by which *mprF* mutation affects the growth rate is unclear, it was reported that growth phase is important in the susceptibility of *S. aureus* toward DAP [28].

Cell wall thickening is also a common feature in DAP-NS *S. aureus* strains [29, 30]. In this study, electro-microscopy revealed that the cell walls of DAP-NS strains were thicker than that of DAP-S strains. Thicker cell walls are considered as a factor contributing to the slower

growth rate. Slower growth in DAP-NS strains may cause resistant bacteria to be overlooked in microbial tests at clinical laboratories.

In conclusion, an SNP in the *mprF* gene led to T345I amino acid substitution and caused up-regulation of positive surface charge, followed by DAP resistance and decreased growth speed. Further investigation is needed to assess the associated mechanisms between this SNP and the bacterial phenotype. Genetic exchange methods can also be applied to other reported SNPs in the *mprF* gene, such as S295L and S337L, to more clearly elucidate the role of *mprF* in DAP resistance.

## Supporting information

**S1 Fig. Alignment of the *mprF* gene in MRSA strains.** The *mprF* sequence of four MRSA strains used in this study were aligned with the reference strain MRSA N315. The red background color indicates base mutations.
(TIF)

**S2 Fig. Population analysis of each strain for DAP susceptibility.** The data represents the additional two independent experiments mentioned in Fig 3.
(TIF)

**S3 Fig. Colony size of each strain.** Each strain was cultured on trypticase soy agar for 24 h at 37˚C. Size bar = 1 cm.
(TIF)

**S4 Fig. Growth curve of each strain.** The data represents the three independent experiments mentioned in Fig 5.
(TIF)

**S5 Fig. Cell wall thickness of each strain.** Cell wall thickness was measured at 200,000× magnification. Fifty cells were included in the calculation of cell wall thickness, and results are expressed as dot plot and mean.
(TIF)

**S1 Table.**
(TIF)

**S2 Table.**
(TIF)

**S3 Table.**
(TIF)

**S4 Table.**
(TIF)

**S1 Raw images.**
(TIF)

## Acknowledgments

We thank Hidehito Matsui for technical assistance with the growth curve method, Tim Foster for providing pIMAY vector, Mayumi Togashi for providing the bacterial strain, Shinsuke Ikeda for helpful advice in the revision process, and Taiji Nakae for useful discussions.

## Author Contributions

**Conceptualization:** Masaki Nakamura, Hideaki Hanaki, Hidero Kitasato.

**Formal analysis:** Masaki Nakamura.

**Investigation:** Masaki Nakamura, Hayato Kawada, Hiroki Uchida, Yusuke Takagi, Shuichi Obata, Ryotaro Eda.

**Methodology:** Masaki Nakamura, Hayato Kawada.

**Project administration:** Hideaki Hanaki, Hidero Kitasato.

**Resources:** Hideaki Hanaki.

**Supervision:** Hideaki Hanaki, Hidero Kitasato.

**Validation:** Masaki Nakamura.

**Visualization:** Masaki Nakamura.

**Writing – original draft:** Masaki Nakamura.

**Writing – review & editing:** Masaki Nakamura, Hayato Kawada, Hidero Kitasato.

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
