## [Decision Letter · Decision Letter 0]

1 Jul 2020

PONE-D-20-09016

Single nucleotide polymorphism leads to daptomycin resistance causing amino acid substitution—T345I in MprF of clinically isolated MRSA strains

PLOS ONE

Dear Dr. Nakamura,

Thank you for submitting your manuscript to PLOS ONE. After careful consideration, we feel that it has merit but does not fully meet PLOS ONE’s publication criteria as it currently stands. Therefore, we invite you to submit a revised version of the manuscript that addresses the points raised during the review process.

We look forward to receiving your revised manuscript.

Kind regards,

Prabagaran Narayanasamy, Ph.D

Academic Editor

PLOS ONE

Journal Requirements:

Additional Editor Comments (if provided):

Recommend to Respond to/correct all the mentioned suggestions.

Reviewers' comments:

Reviewer's Responses to Questions

**Comments to the Author**

1. Is the manuscript technically sound, and do the data support the conclusions?

Reviewer #1: Partly

Reviewer #2: Yes

2. Has the statistical analysis been performed appropriately and rigorously? 

Reviewer #1: Yes

Reviewer #2: Yes

3. Have the authors made all data underlying the findings in their manuscript fully available?

Reviewer #1: Yes

Reviewer #2: Yes

4. Is the manuscript presented in an intelligible fashion and written in standard English?

Reviewer #1: No

Reviewer #2: Yes

5. Review Comments to the Author

Reviewer #1: The authors isolated DAP-resistant and sensitive MRSA strains from a patient and demonstrated that the SNP of mprF gene is responsible for the emergence of DAP-resistant MRSA. Undoubtedly, finding mechanisms of resistance is crucial to develop novel drugs against pathogens. Although detailed mechanism(s) for the development of resistance to daptomycin was not addressed in this manuscript, they performed several experiments, such as a gene exchange and cytochrome c assay, to show that a single mprF gene mutation (T345I) increased resistance to DAP. However, my question is with KAM18. The authors did not address why KAM18 was not used to compare mutant strains (KAM17mut and mutR) in some experiments, such as population analysis, surface charge, and so on. The authors made comparisons by conducting MIC, mprF gene sequence, and POT tests, indicating, I think, that KAM18 is identical or close to KAM17mut. However, there might be other mutations (other than single point mutation in mprF) that cause changes in growth rate and surface charge of KAM18 since several genetic mutations caused by DAP treatment appear to be involved in development of DAP-NS MRSA. It is worth to perform experiments with KAM18 to provide evidence proving that a single mprF gene mutation (T345I) alters the growth rate and surface charge of MRSA. TEM should be performed to measure the cell wall thickness of strains to reveal the relationship between cell wall thickness and the slower growth rate of DAP-NS strain that has T345I mutation in mprF. In addition, TEM study may provide a clue to determine the significance of T345I mutation in altering a cell wall thickness and synthesis. These additional studies will give in significance to the manuscript.

There are additional comments the authors need to address.

1. Line 86, page 4. Need to clarify the statement “ mutation increased (?) DAP susceptibility….”

2. Line 110-111, page 5. Need to correct the comma splice.

3. Line 120, page 6. The footnote “*indicates primers for PCR of mprF” is not clear. Does it mean PCR was done only with these two primers in table 2?

4. Line 211, page 11. Remove parentheses or period.

5. Table 3. It would be better if the full names of drugs are written at the bottom of the table.

6. Line 245, page 13. Need to rewrite sentences. “After cytochrome c bound to the cell and the residual cytochrome c in the supernatant was measured, the adsorbed cytochrome c was calculated.”

7. Line 326-329, page 17. Please rewrite the sentence. It is a long one. It is better to split the sentence.

8. Figure 3 looks the same as Figure S1.

Reviewer #2: Manuscript # PONE-D-20-09016.

The study presented by Nakamura et. al entitled ‘’ Single nucleotide polymorphism……………MRSA isolates is an interesting report, however I have few minor concerns.

1. Generally, the standard abbreviation of the vancomycin is Vanco or VAN, using VCM sounds odd.

2. I would ask to add the Oxacillin MIC of the studied strains to understand so called “see-saw” phenomenon.

3. In line 47: authors are stating that Vancomycin is the oldest antibiotics which is not true. I would suggest not to overemphasize.

4. In continuation of line 52; I would suggest to add the couple of sentences of significance of DAP combination therapy approach for the therapeutic purpose particularly for the prevention of DAP-resistance in MRSA. Combination therapy approach is getting popular for treating complicated infections.

5. Line 65-66: Please add global references as per the statements.

6. Line 70-72: Among them…………………. Phosphatidylglycerol. Add references.

7. The manuscript requires to be reviewed for writing style by English language expert to formulate in better format.

6. PLOS authors have the option to publish the peer review history of their article (what does this mean?). If published, this will include your full peer review and any attached files.

Reviewer #1: No

Reviewer #2: No

---

## [Author Response · Author response to Decision Letter 0]

27 Dec 2020

Thank you for reviewing our manuscript. We revised the manuscript according to your comments and have included an additional experiment. The details of the revised sections are in attached files labeled "Response-to-reviewer1 and 2".

---

## [Decision Letter · Decision Letter 1]

7 Jan 2021

Single nucleotide polymorphism leads to daptomycin resistance causing amino acid substitution—T345I in MprF of clinically isolated MRSA strains

PONE-D-20-09016R1

Dear Dr. Nakamura,

We’re pleased to inform you that your manuscript has been judged scientifically suitable for publication and will be formally accepted for publication once it meets all outstanding technical requirements.

Kind regards,

Prabagaran Narayanasamy, Ph.D

Academic Editor

PLOS ONE

Additional Editor Comments (optional):

Reviewers' comments:

Reviewer's Responses to Questions

**Comments to the Author**

1. If the authors have adequately addressed your comments raised in a previous round of review and you feel that this manuscript is now acceptable for publication, you may indicate that here to bypass the “Comments to the Author” section, enter your conflict of interest statement in the “Confidential to Editor” section, and submit your "Accept" recommendation.

Reviewer #1: All comments have been addressed

Reviewer #2: All comments have been addressed

2. Is the manuscript technically sound, and do the data support the conclusions?

Reviewer #1: Yes

Reviewer #2: Yes

3. Has the statistical analysis been performed appropriately and rigorously? 

Reviewer #1: Yes

Reviewer #2: Yes

4. Have the authors made all data underlying the findings in their manuscript fully available?

Reviewer #1: Yes

Reviewer #2: Yes

5. Is the manuscript presented in an intelligible fashion and written in standard English?

Reviewer #1: Yes

Reviewer #2: Yes

6. Review Comments to the Author

Reviewer #1: (No Response)

Reviewer #2: This manuscript is nice piece of work, however, it does not have enough phenotypic data such as membrane phospholipids to understand the impact of the mprF mutation, fluidity and several others for full length manuscript therefore I would suggest authors to shorten the manuscript as short manuscript.

7. PLOS authors have the option to publish the peer review history of their article (what does this mean?). If published, this will include your full peer review and any attached files.

Reviewer #1: No

Reviewer #2: No

---

## [Editor Report · Acceptance letter]

12 Jan 2021

PONE-D-20-09016R1 

Single nucleotide polymorphism leads to daptomycin resistance causing amino acid substitution—T345I in MprF of clinically isolated MRSA strains 

Dear Dr. Nakamura:

I'm pleased to inform you that your manuscript has been deemed suitable for publication in PLOS ONE. Congratulations! Your manuscript is now with our production department. 

Kind regards, 

on behalf of

Professor Prabagaran Narayanasamy 

Academic Editor

PLOS ONE